# Talin-KANK1 interaction controls the recruitment of cortical microtubule stabilizing complexes to focal adhesions

Benjamin P Bouchet[1], Rosemarie E Gough[2], York-Christoph Ammon[1], Dieudonnée van de Willige[1], Harm Post[3,4,5,6], Guillaume Jacquemet[7], AF Maarten Altelaar[3,4,5,6], Albert JR Heck[3,4,5,6], Benjamin T Goult[2]*, Anna Akhmanova[1]*

[1]Cell Biology, Department of Biology, Faculty of Science, Utrecht University, Utrecht, The Netherlands; [2]School of Biosciences, University of Kent, Canterbury, United Kingdom; [3]Biomolecular Mass Spectrometry and Proteomics, Utrecht University, Utrecht, The Netherlands; [4]Bijvoet Center for Biomolecular Research, Utrecht University, Utrecht, The Netherlands; [5]Utrecht Institute for Pharmaceutical Sciences, Utrecht University, Utrecht, The Netherlands; [6]The Netherlands Proteomics Centre, Utrecht University, Utrecht, The Netherlands; [7]Turku Centre for Biotechnology, University of Turku, Turku, Finland

**Abstract** The cross-talk between dynamic microtubules and integrin-based adhesions to the extracellular matrix plays a crucial role in cell polarity and migration. Microtubules regulate the turnover of adhesion sites, and, in turn, focal adhesions promote the cortical microtubule capture and stabilization in their vicinity, but the underlying mechanism is unknown. Here, we show that cortical microtubule stabilization sites containing CLASPs, KIF21A, LL5β and liprins are recruited to focal adhesions by the adaptor protein KANK1, which directly interacts with the major adhesion component, talin. Structural studies showed that the conserved KN domain in KANK1 binds to the talin rod domain R7. Perturbation of this interaction, including a single point mutation in talin, which disrupts KANK1 binding but not the talin function in adhesion, abrogates the association of microtubule-stabilizing complexes with focal adhesions. We propose that the talin-KANK1 interaction links the two macromolecular assemblies that control cortical attachment of actin fibers and microtubules.

*For correspondence: B.T.Goult@kent.ac.uk (BTG); a.akhmanova@uu.nl (AA)

## Introduction

Cell adhesions to the extracellular matrix support epithelial integrity and cell migration, and also provide signaling hubs that coordinate cell proliferation and survival (*Hynes, 1992*). Integrin-based adhesions (focal adhesions, FAs) are large macromolecular assemblies, in which the cytoplasmic tails of integrins are connected to the actin cytoskeleton. One of the major components of FAs is talin, a ~2500 amino acid dimeric protein, which plays a key role in adhesion formation by activating integrins (*Anthis et al., 2009*), coupling them to cytoskeletal actin (*Atherton et al., 2015*), regulating adhesion dynamics and recruiting different structural and signaling molecules (*Calderwood et al., 2013*; *Gardel et al., 2010*; *Wehrle-Haller, 2012*).

While the major cytoskeletal element associated with FAs is actin, microtubules also play an important role in adhesion by regulating the FA turnover (*Akhmanova et al., 2009*; *Byron et al., 2015*; *Kaverina et al., 1999*, *1998*; *Krylyshkina et al., 2003*; *Small and Kaverina, 2003*; *Stehbens and Wittmann, 2012*; *Yue et al., 2014*). The recent proteomics work showed that

**eLife digest** Animal cells are organized into tissues and organs. A scaffold-like framework outside of the cells called the extracellular matrix provides support to the cells and helps to hold them in place. Cells attach to the extracellular matrix via structures called focal adhesions on the cell surface; these structures contain a protein called talin.

For a cell to be able to move, the existing focal adhesions must be broken down and new adhesions allowed to form. This process is regulated by the delivery and removal of different materials along fibers called microtubules. Microtubules can usually grow and shrink rapidly, but near focal adhesions they are captured at the surface of the cell and become more stable. However, it is not clear how focal adhesions promote microtubule capture and stability.

Bouchet et al. found that a protein called KANK1 binds to the focal adhesion protein talin in human cells grown in a culture dish. This allows KANK1 to recruit microtubules to the cell surface around the focal adhesions by binding to particular proteins that are associated with microtubules. Disrupting the interaction between KANK1 and talin by making small alterations in these two proteins blocked the ability of focal adhesions to capture surrounding microtubules. The next step following on from this work will be to find out whether this process also takes place in the cells within an animal's body, such as a fly or a mouse.

microtubule-FA cross-talk strongly depends on the activation state of the integrins (*Byron et al., 2015*). Microtubules can affect adhesions by serving as tracks for delivery of exocytotic carriers (*Stehbens et al., 2014*), by controlling endocytosis required for adhesion disassembly (*Ezratty et al., 2005*; *Theisen et al., 2012*) and by regulating the local activity of signaling molecules such as Rho GTPases (for review, see [*Kaverina and Straube, 2011*; *Stehbens and Wittmann, 2012*]).

In contrast to actin, which is directly coupled to FAs, microtubules interact with the plasma membrane sites that surround FAs. A number of proteins have been implicated in microtubule attachment and stabilization in the vicinity of FAs. Among them are the microtubule plus end tracking proteins (+TIPs) CLASP1/2 and the spectraplakin MACF1/ACF7, which are targeted to microtubule tips by EB1, and a homologue of EB1, EB2, which binds to mitogen-activated protein kinase kinase kinase kinase 4 (MAP4K4) (*Drabek et al., 2006*; *Honnappa et al., 2009*; *Kodama et al., 2003*; *Mimori-Kiyosue et al., 2005*). The interaction of CLASPs with the cell cortex depends on the phosphatidylinositol 3, 4, 5-trisphosphate (PIP3)-interacting protein LL5β, to which CLASPs bind directly, and is partly regulated by PI-3 kinase activity (*Lansbergen et al., 2006*). Other components of the same cortical assembly are the scaffolding proteins liprin-α1 and β1, a coiled-coil adaptor ELKS/ERC1, and the kinesin-4 KIF21A (*Lansbergen et al., 2006*; *van der Vaart et al., 2013*). Both liprins and ELKS are best known for their role in organizing presynaptic secretory sites (*Hida and Ohtsuka, 2010*; *Spangler and Hoogenraad, 2007*); in agreement with this function, ELKS is required for efficient constitutive exocytosis in HeLa cells (*Grigoriev et al., 2007*, *2011*). LL5β, liprins and ELKS form micrometer-sized cortical patch-like structures, which will be termed here cortical microtubule stabilization complexes, or CMSCs. The CMSCs are strongly enriched at the leading cell edges, where they localize in close proximity of FAs but do not overlap with them ([*Lansbergen et al., 2006*; *van der Vaart et al., 2013*], reviewed in [*Astro and de Curtis, 2015*]). They represent a subclass of the previously defined plasma membrane-associated platforms (PMAPs) (*Astro and de Curtis, 2015*), which have overlapping components such as liprins, but may not be necessarily involved in microtubule regulation, as is the case for liprin-ELKS complexes in neurons, where they are part of cytomatrix at the active zone (*Gundelfinger and Fejtova, 2012*).

Several lines of evidence support the importance of the CMSC-FA cross-talk. In migrating keratinocytes, LL5β and CLASPs accumulate around FAs and promote their disassembly by targeting the exocytosis of matrix metalloproteases to FA vicinity (*Stehbens et al., 2014*). Furthermore, liprin-α1, LL5α/β and ELKS localize to protrusions of human breast cancer cells and are required for efficient cell migration and FA turnover (*Astro et al., 2014*). In polarized epithelial cells, LL5β and CLASPs are found in the proximity of the basal membrane, and this localization is controlled by the integrin

activation state (*Hotta et al., 2010*). CLASP and LL5-mediated anchoring of MTs to the basal cortex also plays a role during chicken embryonic development, where it prevents the epithelial-mesenchymal transition of epiblast cells (*Nakaya et al., 2013*). LL5β, CLASPs and ELKS were also shown to concentrate at podosomes, actin-rich structures, which can remodel the extracellular matrix (*Proszynski and Sanes, 2013*). Interestingly, LL5β-containing podosome-like structures are also formed at neuromuscular junctions (*Kishi et al., 2005*; *Proszynski et al., 2009*; *Proszynski and Sanes, 2013*), and the complexes of LL5β and CLASPs were shown to capture microtubule plus ends and promote delivery of acetylcholine receptors (*Basu et al., 2015*, *2014*; *Schmidt et al., 2012*).

While the roles of CMSCs in migrating cells and in tissues are becoming increasingly clear, the mechanism underlying their specific targeting to integrin adhesion sites remains elusive. Recently, we found that liprin-β1 interacts with KANK1 (*van der Vaart et al., 2013*), one of the four members of the KANK family of proteins, which were proposed to act as tumor suppressors and regulators of cell polarity and migration through Rho GTPase signaling (*Gee et al., 2015*; *Kakinuma et al., 2008*, *2009*; *Li et al., 2011*; *Roy et al., 2009*). KANK1 recruits the kinesin-4 KIF21A to CMSCs, which inhibits microtubule polymerization and prevents microtubule overgrowth at the cell edge (*Kakinuma and Kiyama, 2009*; *van der Vaart et al., 2013*). Furthermore, KANK1 participates in clustering of the other CMSC components (*van der Vaart et al., 2013*).

Here, we found that KANK1 is required for the association of the CMSCs with FAs. The association of KANK1 with FAs depends on the KN domain, a conserved 30 amino acid polypeptide sequence present in the N-termini of all KANK proteins. Biochemical and structural analysis showed that the KN domain interacts with the R7 region of the talin rod. Perturbation of this interaction both from the KANK1 and the talin1 side prevented the accumulation of CMSC complexes around focal adhesions and affected microtubule organization around FAs. We propose that KANK1 molecules, recruited by talin to the outer rims of FA, serve as 'seeds' for organizing other CMSC components in the FA vicinity through multivalent interactions between these components. This leads to co-organization of two distinct cortical assemblies, FAs and CMSCs, responsible for the attachment of actin and microtubules, respectively, and ensures effective cross-talk between the two types of cytoskeletal elements.

## Results

### Identification of talin1 as a KANK1 binding partner

Our previous work showed that the endogenous KANK1 colocalizes with LL5β, liprins and KIF21A in cortical patches that are closely apposed to, but do not overlap with FAs (*van der Vaart et al., 2013*). We confirmed these results both in HeLa cells and the HaCaT immortal keratinocyte cell line, in which CMSC components CLASPs and LL5β were previously shown to strongly cluster around FAs and regulate their turnover during cell migration (*Stehbens et al., 2014*) (*Figure 1—figure supplement 1A,B*). Inhibition of myosin-II with blebbistatin, which reduces tension on the actin fibers and affects the activation state of FA molecules, such as integrins and talin (*Parsons et al., 2010*), caused not only FA disassembly but also a strong reduction in clustering of CMSC components at the cell periphery (*Figure 1—figure supplement 2A,B*), as described previously (*Stehbens et al., 2014*). To investigate this effect in more detail, we partially inhibited contractility using a Rho-associated protein kinase 1 (ROCK1) inhibitor, Y-27632 (*Oakes et al., 2012*). In these conditions, the number of FAs was not affected although their size was reduced (*Figure 1—figure supplement 2C–E*). This treatment was sufficient to diminish CMSC clustering at the cell edge (*Figure 1—figure supplement 2C,F*). Interestingly, at the same time we observed a very significant increase in the overlap of KANK1 with FA adhesion markers (*Figure 1—figure supplement 2C,G*). Live imaging of KANK1 together with the FA marker paxillin showed a gradual redistribution of KANK1 into the areas occupied by FAs upon ROCK1 inhibitor-induced attenuation of contractility (*Figure 1—figure supplement 2H*, *Video 1*). These data indicate that the organization of CMSCs at the cell cortex might be controlled by interactions with tension-sensitive components of FAs.

To identify the domains of KANK1 required for cortical localization, we performed deletion mapping. KANK1 comprises an N-terminal KANK family-specific domain of unknown function, the KN domain (residues 30–68) (*Kakinuma et al., 2009*), a coiled coil region, the N-terminal part of which interacts with liprin-β1, and a C-terminal ankyrin repeat domain, which binds to KIF21A (*van der*

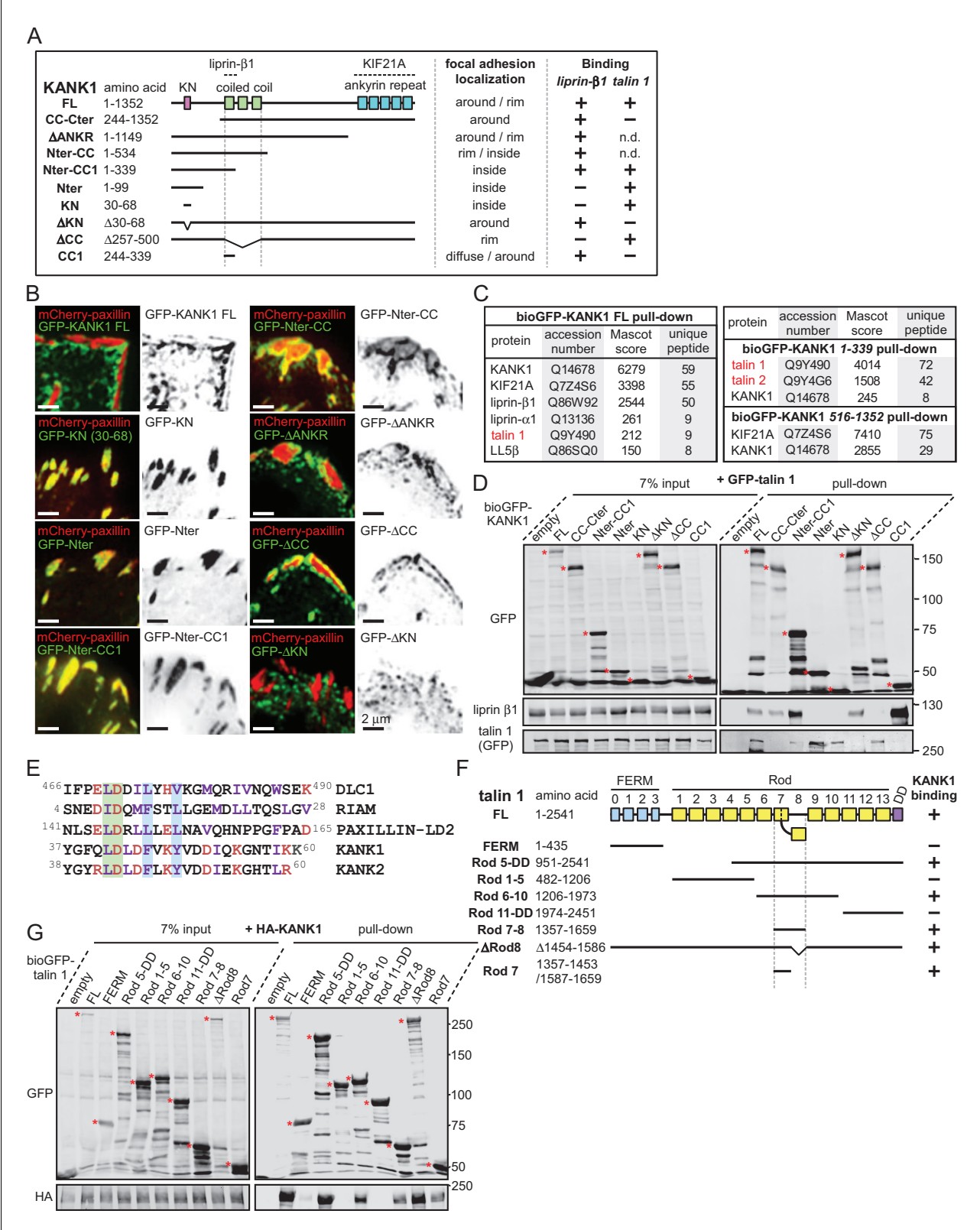

**Figure 1.** The KN motif of KANK1 interacts with the R7 domain of talin1. (**A**) Schematic representation of KANK1 and the deletion mutants used in this study, and the summary of their interactions and localization. N.d., not determined in this study. (**B**) TIRFM images of live HeLa cells transiently expressing the indicated GFP-tagged KANK1 deletion mutants together with the focal adhesion marker mCherry-paxillin. In these experiments, endogenous KANK1 and KANK2 were also expressed. (**C**) Identification of the binding partners of Bio-GFP-tagged KANK1 and its indicated deletion

*Figure 1 continued on next page*

*Figure 1 continued*

mutants by using streptavidin pull down assays from HEK293T cells combined with mass spectrometry. (**D**) Streptavidin pull down assays with the BioGFP-tagged KANK1 or the indicated KANK1 mutants, co-expressed with GFP-talin1 in HEK293T cells, analyzed by Western blotting with the indicated antibodies. (**E**) Sequence alignment of KANK1 and KANK2 with the known talin-binding sites of DLC1, RIAM and Paxillin. The LD-motif and the interacting hydrophobic residues are highlighted green and blue respectively. (**F**) Schematic representation of talin1 and the deletion mutants used in this study, and their interaction with KANK1. (**G**) Streptavidin pull down assays with the BioGFP-tagged talin1 or the indicated talin1 mutants, co-expressed with HA-KANK1 in HEK293T cells, analyzed by Western blotting with the indicated antibodies.

The following source data and figure supplements are available for figure 1:

**Figure supplement 1.** KANK1 colocalizes with CMSC components around FAs.

**Figure supplement 2.** Role of myosin II activity in KANK1 localization to FA.

**Figure supplement 2—source data 1.** An Excel sheet with numerical data on the quantification of peripheral clustering of different markers, FA number and area and colocalization of KANK1 with talin represented as plots in *Figure 1—figure supplement 2B,D–G*.

**Figure supplement 3.** FA localization of KN-bearing proteins.

*Vaart et al., 2013*), while the rest of the protein is predicted to be unstructured (*Figure 1A*). Surprisingly, the KN domain alone strongly and specifically accumulated within FAs (*Figure 1B*). A similar localization was also seen with a somewhat larger N-terminal fragment of KANK1, Nter, as well as the Nter-CC1 deletion mutant, which contained the first, liprin-β1-binding coiled coil region of KANK1 (*Figure 1A,B*). However, an even larger N-terminal part of KANK1, encompassing the whole coiled coil domain (Nter-CC) showed a pronounced enrichment at the FA rim (*Figure 1A,B*). The KANK1 deletion mutant missing only the C-terminal ankyrin repeat domain (△ANKR) was completely excluded from FAs but accumulated in their immediate vicinity, similar to the full-length KANK1 (*Figure 1A,B*). A tight ring-like localization at the outer rim of FAs was also observed with a KANK1 mutant, which completely missed the coiled coil region but contained the ankyrin repeat domain (△CC), while the mutant which missed just the KN domain showed no accumulation around FAs (*Figure 1A,B*). To test whether the exclusion of larger KANK1 fragments from the FA core was simply due to the protein size, we fused GFP-tagged KN domain to the bacterial protein β-D-galactosidase (LacZ), but found that this fusion accumulated inside and not around FAs (*Figure 1— figure supplement 3*). Since GFP-KN-LacZ and GFP-KANK1-△ANKRD have a similar size (1336 and 1400 amino acids, respectively), but one accumulates inside FAs, while the other is excluded to their periphery, this result suggests that features other than the mere protein size determine the specific localization of KANK1 to the FA rim. We conclude that the KN domain of KANK1 has an affinity for FAs, but the presence of additional KANK1 sequences prevents the accumulation of the protein inside FAs and instead leads to the accumulation of KANK1 at the FA periphery.

To identify the potential FA-associated partners of KANK1, we co-expressed either full-length KANK1 or its N-terminal and C-terminal fragments fused to GFP and a biotinylation (Bio) tag together with biotin ligase BirA in HEK293T cells and performed streptavidin pull down assays combined with mass spectrometry. In addition to

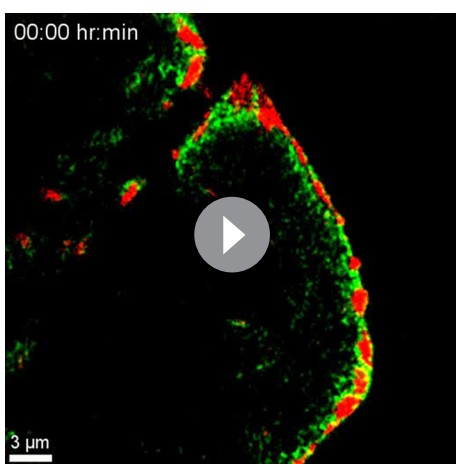

**Video 1.** Effect of myosin II inhibition on KANK1 localization to FA. TIRFM-based time-lapse imaging of HeLa cells stably expressing GFP-KANK1 and TagRFP-paxillin and treated when indicated with 10 µM ROCK1 inhibitor Y-27632. Both red and green fluorescence images were acquired at 1 min interval and displayed at 15 frames/second (accelerated 900 times).

the already known binding partners of KANK1, such as KIF21A, liprins and LL5β, we identified talin1 among the strongest hits (*Figure 1C*). Talin2 was also detected in a pull down with the KANK1 N-terminus though not with the full-length protein (*Figure 1C*). The interaction between KANK1 and talin1 was confirmed by Western blotting, and subsequent deletion mapping showed that the talin1-binding region of KANK1 encompasses the KN domain (*Figure 1A,D*), while liprin-β1 binds to the N-terminal part of the coiled coil domain, as shown previously (*van der Vaart et al., 2013*).

Sequence analysis of the KN domain showed that it is predicted to form a helix and contains a completely conserved leucine aspartic acid-motif (LD-motif) (*Alam et al., 2014*; *Zacharchenko et al., 2016*). The LD-motifs in RIAM (*Goult et al., 2013*), DLC1 and Paxillin (*Zacharchenko et al., 2016*) have recently been identified as talin-binding sites that all interact with talin via a helix addition mechanism. Alignment of the KN domain of KANK with the LD-motif of DLC1, RIAM and Paxillin (*Zacharchenko et al., 2016*) revealed that the hydrophobic residues that mediate interaction with talin are present in the KN domain (*Figure 1E*).

Using deletion analysis, we mapped the KANK1-binding site of talin1 to the central region of the talin rod, comprising the R7-R8 domains (*Figure 1F*). This R7-R8 region of talin is unique (*Gingras et al., 2010*), as the 4-helix bundle R8 is inserted into a loop of the 5-helix bundle R7, and thus protrudes from the linear chain of 5-helix bundles of the talin rod (*Figures 1F*, *2A*). This R8 domain serves as a binding hub for numerous proteins including vinculin, synemin and actin (*Calderwood et al., 2013*). R8 also contains a prototypic recognition site for LD-motif proteins, including DLC1 (*Figure 2B*), Paxillin and RIAM (*Zacharchenko et al., 2016*). Based on the presence of the LD-binding site, we anticipated that KANK1 would also interact with R8. However, deletion mapping revealed that KANK1 in fact binds to the talin1 rod domain R7 (*Figure 1F,G*), suggesting that KANK1 interacts with a novel binding site on talin1.

## Structural characterization and mutational analysis of the KANK1-talin1 complex

To explore the interaction between talin1 and KANK1 in more detail, we used NMR chemical shift mapping using $^{15}$N-labeled talin1 R7-R8 (residues 1357–1653) and a synthetic KANK1 peptide of the KN domain, KANK1(30–60). The addition of the KANK1(30–60) peptide resulted in large spectral changes (*Figure 2C*), most of which were in the slow exchange regime on the NMR timescale indicative of a tight interaction. In agreement with the pull down data, the signals that shifted in slow exchange upon the addition of KANK1(30–60) mapped largely onto the R7 domain with only small progressive shift changes evident on R8. To validate R7 as the major KANK1-binding site on talin, we repeated the NMR experiments using the individual domains, R8 (residues 1461–1580) and R7 (residues 1359–1659 Δ1454–1586). Addition of KANK1(30–60) induced small chemical shift changes on the R8 domain indicative of a weak interaction (most likely due to the interaction of LD with the LD-recognition box on R8). However, the addition of a 0.5 molar ratio of KANK1(30–60) to R7 induced large spectral changes with many of the peaks displaying two locations, corresponding to the free peak position and the bound peak position. This is indicative of slow-exchange and confirms a high affinity interaction between R7 and KANK1. The KN peptide is the first identified ligand for the R7 domain.

NMR chemical shifts also provide information on the residues involved in the interaction, as the peaks in the $^{15}$N-HSQC spectrum pertain to individual residues in the protein. To map these chemical shift changes onto the structure of R7-R8, it was first necessary to complete the backbone chemical shift assignments of the R7 domain. This was achieved using conventional triple resonance experiments as described previously (*Banno et al., 2012*), using $^{13}$C,$^{15}$N labeled R7. The chemical shift changes mapped predominantly onto one face of R7, comprised of helices 2 and 5 of the 5-helix bundle (talin rod helices 29 and 36), as shown in *Figure 2D–E*.

Our recent elucidation of the interaction between the LD-motif of DLC1 and talin R8 has generated insight into how LD-motifs are recognized by helical bundles (PDB ID. 5FZT, [*Zacharchenko et al., 2016*]). In the DLC1:talin R8 complex the DLC1 peptide adopts a helical conformation that packs against two helices on the side of the helical bundle. It is becoming increasingly clear that other LD-motif proteins bind to talin through a similar interaction mode (*Zacharchenko et al., 2016*). The surface of α2 and α5 on R7 forms a hydrophobic groove that the KANK1 helix docks into. A striking feature of this KANK1 binding surface is that the two helices are held apart by the conserved aromatic residues, W1630 at the end of α5 and Y1389 at the end of α2

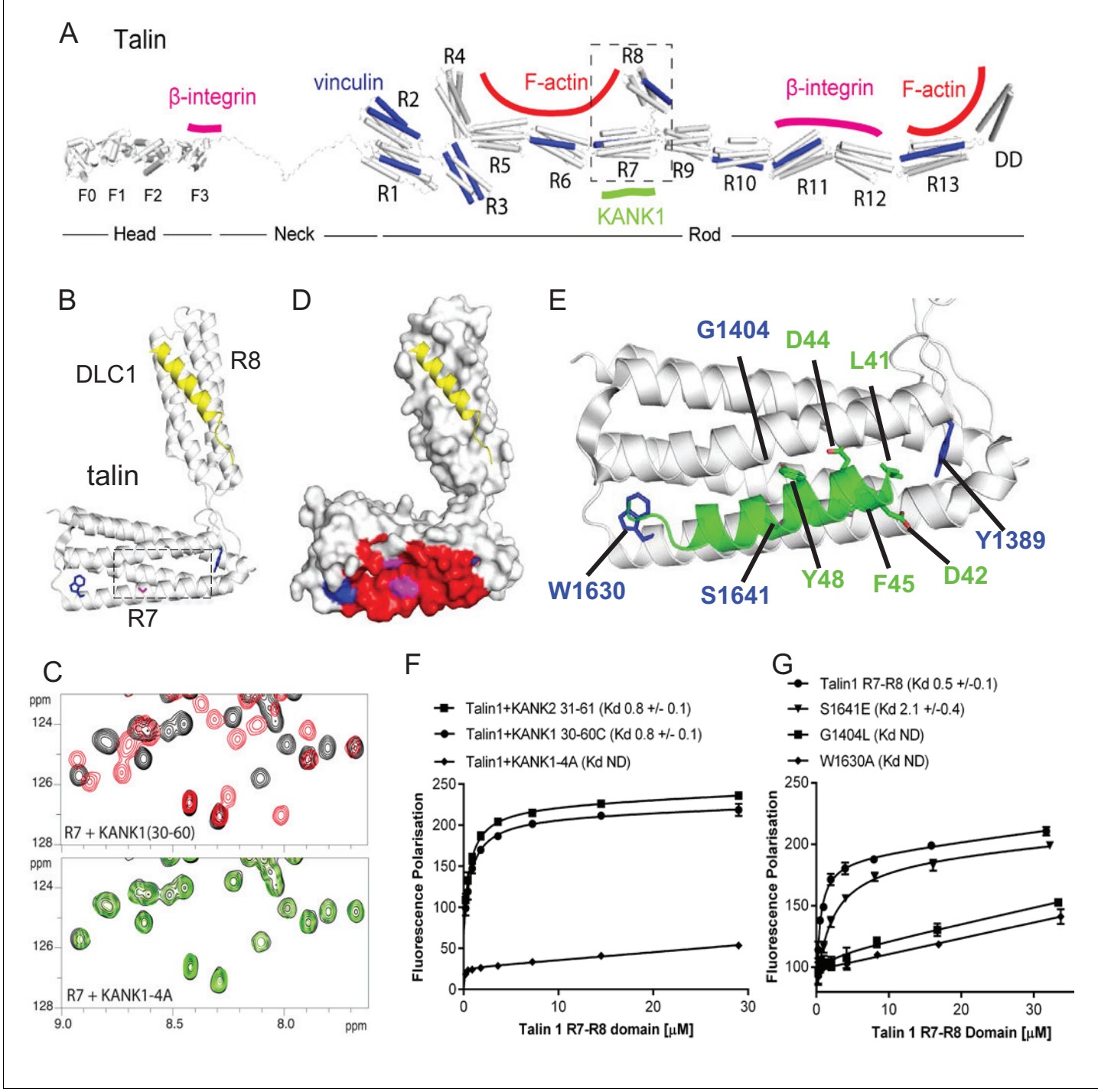

**Figure 2.** Biochemical and structural characterization of the Talin-KANK interaction. (**A**) Schematic representation of Talin1, with F-actin, β-integrin and vinculin binding sites highlighted. The KANK1 binding site on R7 is also shown. (**B**) The structure of the complex between talin1 R7-R8 (white) and the LD-motif of DLC1 (yellow) bound on the R8 subdomain (PDB ID. 5FZT, [*Zacharchenko et al., 2016*]). Residues W1630 and Y1389 (blue) and S1641 (magenta) are highlighted. (**C–D**) The KANK KN domain binds to a novel site on talin R7. $^1$H,$^{15}$N HSQC spectra of 150 µM $^{15}$N-labelled talin1 R7 (residues 1357–1659 Δ1454–1586) in the absence (black) or presence of KANK1(30–68)C peptide (red) (top panel) or KANK1-4A (green) (bottom panel) at a ratio of 1:3. (**D**) Mapping of the KANK1 binding site on R7 as detected by NMR using weighted chemical shift differences (red) – mapped onto the R7-R8 structure in (**B**). Residues W1630 and Y1389 (blue) and G1404 and S1641 (magenta) are highlighted. (**E**) Structural model of the talin1:KANK1 interface. Ribbon representation of the KANK1 binding site, comprised of the hydrophobic groove between helices 29 and 36 of R7. Two bulky conserved residues, W1630 and Y1389 (blue) hold these two helices apart forming the binding interface. A small glycine side chain (G1404) creates a pocket between the helices. S1641 (magenta) has been shown previously to be a phosphorylation site (*Ratnikov et al., 2005*). The KN peptide (green)

*Figure 2 continued on next page*

*Figure 2 continued*

docked onto the KANK binding surface. (F–G) Biochemical characterization of the talin:KANK interaction. (F) Binding of BODIPY-labeled KANK1(30–60) C, KANK2(31–61)C and KANK1-4A peptides to Talin1 R7-R8 (1357–1659) was measured using a Fluorescence Polarization assay. (G) Binding of BODIPY-labeled KANK1(30–60)C to wild type R7-R8, R7-R8 S1641E, R7-R8 G1404L and R7-R8 W1630A. Dissociation constants ± SE (µM) for the interactions are indicated in the legend. All measurements were performed in triplicate. ND, not determined.

The following figure supplements are available for figure 2:

**Figure supplement 1.** NMR validation of the Talin1 R7-R8 mutants.

**Figure supplement 2.** Biochemical characterization of the Talin2:KANK interaction.

(*Figure 2B,E*). W1630 and Y1389 thus essentially serve as molecular rulers, separating helices α2 and α5 by ~8Å (compared with ~5–6Å for the other bundles in R7-R8). The spacing between the two helices is enhanced further as the residues on the inner helical faces, S1400, G1404, S1411 on α2 and S1637 and S1641 on α5, have small side chains which have the effect of creating two conserved pockets midway along the hydrophobic groove of the KANK1-binding site (*Figure 2E*).

The talin-binding site on KANK1 is unusual as it contains a double LD-motif, LDLD. The structure of R7 revealed a potential LD-recognition box with the positive charges, K1401 and R1652 positioned on either side to engage either one, or both, of the aspartic residues. Using the docking program HADDOCK (*van Zundert et al., 2016*), we sought to generate a structural model of the KANK1/R7 complex, using the chemical shift mapping on R7 and a model of KANK1(30–60) as a helix (created by threading the KANK1 sequence onto the DLC1 LD-motif helix). This analysis indicated that the KANK-LD helix can indeed pack against the sides of α2 and α5 (*Figure 2E*). Interestingly, all of the models, irrespective of which way the KANK1 helix ran along the surface, positioned the bulky aromatic residue, Y48 in KANK1, in the hydrophobic pocket created by G1404. This raised the possibility that mutation of G1404 to a bulky hydrophobic residue might block KANK1 binding by preventing Y48 engagement. We also noticed that S1641, one of the small residues that create the pocket, has been shown to be phosphorylated in vivo (*Ratnikov et al., 2005*) and might have a regulatory function in the KANK1-talin1 interaction.

To test these hypotheses, we generated a series of point mutants in talin R7 and also in the KANK1 KN-domain, designed to disrupt the talinR7/KANK1 interaction. On the KANK1 side, we mutated the LDLD motif to AAAA, (the KANK1-4A mutant), while on the talin1 side, we generated a series of R7 mutants. These included G1404L, in which a bulky hydrophobic residue was introduced instead of glycine to occlude the hydrophobic pocket in R7, S1641E, a phosphomimetic mutant aimed to test the role of talin phosphorylation in regulating KANK1 binding, and W1630A, a substitution that would remove one of the molecular rulers holding α2 and α5 helices apart at a fixed distance. These mutants were introduced into talin1 R7-R8 and the structural integrity of the mutated proteins confirmed using NMR (*Figure 2—figure supplement 1*). The relative binding affinities were measured using an in vitro fluorescence polarization assay. In this assay, the KANK1(30–60) peptide is fluorescently labeled with BODIPY and titrated with an increasing concentration of talin R7-R8, and the binding between the two polypeptides results in an increase in the fluorescence polarization signal (*Figure 2F*). The KANK1-4A mutant abolished binding to talin (*Figure 2C,F*). The S1641E mutant had only a small effect on binding (*Figure 2G*), suggesting that either talin1 phosphorylation does not play a major role in modulating the interaction with KANK1 or that the S-E mutation is not a good phosphomimetic, possibly because phosphorylation might also affect helix formation integrity, an effect not mimicked by a single aspartate residue. However, strikingly, both the W1630A and the G1404L mutants abolished binding of KANK1 to talin R7 (*Figure 2G*), confirming the validity of our model. Finally, we also tested whether the KN-R7 interaction is conserved in talin2 and KANK2, and found that this was indeed the case (*Figure 2—figure supplement 2*). We conclude that the conserved KN domain of KANKs is a talin-binding site.

## Talin1-KANK1 interaction controls cortical organization of CMSC components

Next, we set out to test the importance of the identified interactions in a cellular context by using the KANK1-4A and the talin G1404L mutants. We chose the G1404L talin mutant over W1630A for our cellular studies, because removing the bulky tryptophan from the hydrophobic core of the R7 might have the off target effect of perturbing the mechanical stability of R7, and our recent studies showed that the mechanostability of R7 is important for protecting R8 from force-induced talin extension (*Yao et al., 2016*). As could be expected based on the binding experiments with purified protein fragments, the 4A mutation reduced the interaction of the full-length KANK1 with talin1 in a pull-down assay (*Figure 3A*). An even stronger reduction was observed when KANK-△CC or the KN alone were tested (*Figure 3A*). Furthermore, the introduction of the G1404L mutation abrogated the interaction of full-length talin1 or its R7 fragment with full-length KANK1 (*Figure 3B*).

To investigate the localization of KANK1-4A, we used HeLa cells depleted of endogenous KANK1 and KANK2, the two KANK isoforms present in these cells based on our proteomics studies (*van der Vaart et al., 2013*) (*Figure 3—figure supplement 1A*), in order to exclude the potential oligomerization of the mutant KANK1 with the endogenous proteins. Rescue experiments were performed using GFP-KANK1, resistant for the used siRNAs due to several silent mutations (*van der Vaart et al., 2013*), or its 4A mutant. We also included in this analysis a KANK1 mutant lacking the liprin-β 1-binding coiled coil domain (the △CC deletion mutant, *Figure 1A*), and the 4A-version of the KANK1-△CC deletion mutant. Total Internal Reflection Fluorescence Microscopy (TIRFM)-based live imaging showed that, consistent with our previous results, the GFP-tagged wild type KANK1 strongly accumulated in cortical patches that were tightly clustered around FAs (*Figure 3C,D*). The KANK1-△CC mutant, which lacked the liprin-β1-binding site but contained an intact KN motif, showed highly specific ring-like accumulations at the rims of FAs (*Figure 3C,D*). In contrast, KANK1-4A was not clustered anymore around FAs but was dispersed over the cell cortex (*Figure 3C,D*). The KANK1-△CC-4A mutant, lacking both the liprin-β1 and the talin-binding sites, and the KN-4A mutant were completely diffuse (*Figure 3C,D*).

To test the impact of the talin1-G1404L mutant, we depleted both talin1 and talin2, which are co-expressed in HeLa cells (*Figure 3—figure supplement 1B*), and rescued them by introducing mouse GFP-talin1, which was resistant to used siRNAs. The depletion of the two talin proteins resulted in a dramatic loss of FAs and cell detachment from coverslips (data not shown), in agreement with the well-established essential role of talin1 in FA formation (*Calderwood et al., 2013*; *del Rio et al., 2009*; *Yan et al., 2015*; *Yao et al., 2014*). Therefore, in this experiment only cells expressing GFP-talin1 displayed normal attachment and spreading (*Figure 3—figure supplement 1C*). The GFP-talin1-G1404L mutant could fully support cell attachment and spreading, although the cell area was slightly increased compared to cells rescued with the wild type GFP-talin1 (*Figure 3—figure supplement 1C–E*). The number and size of focal adhesions were not significantly different between the cells rescued with the wild type talin1 or its G1404L mutant (*Figure 3—figure supplement 1F,G*), indicating that the mutant is functional in supporting FA formation. Strikingly, while in cells expressing the wild-type talin1, KANK1 was prominently clustered around FAs, it was dispersed over the plasma membrane in cells expressing talin1-G1404L (*Figure 3E,F*, *Figure 1—figure supplement 1*). We conclude that perturbing the KANK1-talin interaction, including the use of a single point mutation in the ~2500 amino acid long talin1 protein, which does not interfere with the talin function in FA formation, abrogates KANK1 association with FAs.

We next tested whether mislocalization of KANK1 due to the perturbation of KANK1-talin1 binding affected other CMSC components. The localization of GFP-KANK1 and its mutants relative to FAs labeled with endogenous markers was very similar to that described above based on live imaging experiments (*Figure 4A*). Co-depletion of KANK1 and KANK2 abolished clustering of CMSC components, such as LL5β and KIF21A at the cell edge (*Figure 4B,C*). Wild type GFP-KANK1 could rescue cortical clustering of these proteins in KANK1 and KANK2-depleted cells (*Figure 4B,C*). However, this was not the case for the KANK1-4A mutant, the KANK1-△CC mutant or the KANK1 version bearing both mutations (*Figure 4B,C*). Importantly, the dispersed puncta of the KANK1-4A mutant still colocalized with LL5β, as the binding to liprin-β1 was intact in this mutant (*Figure 3A*, *Figure 4B,C*), while the FA-associated rings of KANK1-△CC, the mutant deficient in liprin-β1 binding, showed a mutually exclusive localization with LL5β (*Figure 4B*). In contrast, KIF21A, which binds

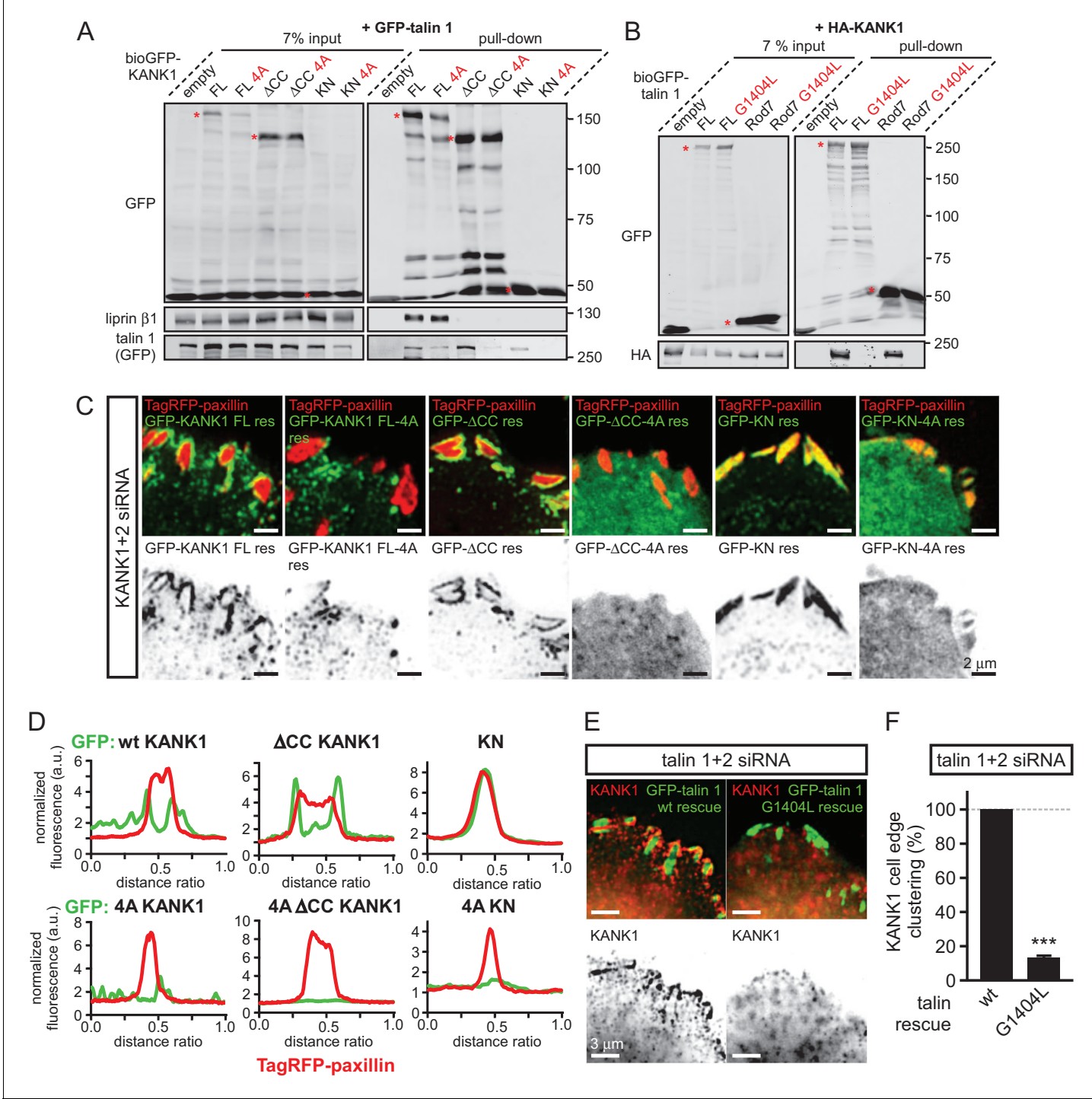

**Figure 3.** KANK1-talin interaction is required for recruiting KANK1 to FAs. (A) Streptavidin pull-down assays with the BioGFP-tagged KANK1 or the indicated KANK1 mutants, co-expressed with GFP-talin1 in HEK293T cells, analyzed by Western blotting with the indicated antibodies. (B) Streptavidin pull down assays with the BioGFP-tagged talin1 or the indicated talin1 mutants, co-expressed with HA-KANK1 in HEK293T cells, analyzed by Western blotting with the indicated antibodies. (C) TIRFM images of live HeLa cells depleted of KANK1 and KANK2 and co-expressing the indicated siRNA-resistant GFP-KANK1 fusions and TagRFP-paxillin. (D) Fluorescence profile of GFP-tagged mutants and TagRFP-paxillin based on line scan measurement across the FA area in TIRFM images as in panel (C). (E) Widefield fluorescence images of HeLa cells depleted of endogenous talin1 and talin2, rescued by the expression of the wild type GFP-tagged mouse talin1 or its G1404L mutant and labeled for endogenous KANK1 by immunofluorescence staining. (F) Quantification of peripheral clustering of KANK1 in cells treated and analyzed as in (E) (n=12, 6 cells per condition). Error bar, SEM; ***p<0.001, Mann-Whitney U test.

*Figure 3 continued on next page*

*Figure 3 continued*

The following source data and figure supplements are available for figure 3:

**Source data 1.** An Excel sheet with numerical data on the quantification of peripheral clustering of KANK1 represented as a plot in *Figure 3F*.

**Figure supplement 1.** Validation of KANK1/2 and talin1/2 knockdown and effect of disrupted KANK/talin 1 binding in cell spreading and FA formation in HeLa cells.

**Figure supplement 1—source data 1.** An Excel sheet with numerical data on the quantification of cell area, FA number and FA area represented as plots in *Figure 3—figure supplement 1E–G*.

to the ankyrin repeat domain of KANK1, could still colocalize with KANK1-△CC at FA rims (*Figure 4B*). The overall accumulation of KIF21A at the cell periphery was, however, reduced, in line with the strongly reduced KANK1 peripheral clusters observed with the KANK1-△CC mutant. The diffuse localization of the KANK1-4A-△CC mutant led to the strongly dispersed distribution of the CMSC markers (*Figure 4B,C*). Furthermore, only the full-length wild type KANK1, but neither the 4A nor △CC mutant could support efficient accumulation of CLASP2 at the peripheral cell cortex in KANK1 and KANK2-depleted cells (*Figure 4D,E*), in line with the fact that cortical recruitment of CLASPs depends on LL5β (*Lansbergen et al., 2006*).

Next, we investigated whether disrupting the KANK1-talin1 interaction from the talin1 side would affect also CMSC localization and found that this was indeed the case: both LL5β and KIF21A were clustered around FAs in talin1 and talin2-depleted cells rescued with the wild type GFP-talin1, but not in the cells expressing the GFP-talin1-G1404L mutant, deficient in KANK1 binding (*Figure 4F,G*).

Our data showed that KANK1-△CC could not support proper clustering of CMSC components at the cell edge in spite of its tight accumulation at the FA rims. These data indicate that in addition to binding to talin1, the localization of CMSC clusters depends on the KANK1-liprin-β1 connection. This notion is supported by the observation that the overexpressed coiled coil region of KANK1 (CC1), which can compete for liprin-β1 binding but cannot interact with talin1, acted as a very potent dominant negative, which suppressed accumulation of LL5β at the cell periphery (*Figure 4H,I*). We conclude that the core CMSC protein LL5β as well as the microtubule-binding CMSC components KIF21A and CLASP2 depend on the KANK1 interaction with both talin1 and liprin-β1 for their efficient clustering in the vicinity of focal adhesions at the cell periphery.

## Disruption of KANK1-talin1 binding perturbs microtubule plus end organization at the cell periphery

We next investigated the impact of the disruption of KANK1-talin1 interaction on microtubule organization. Due to their stereotypic round shape, HeLa cells represent a particularly convenient model for studying the impact of CMSCs on the distribution and dynamics of microtubule plus ends (*Lansbergen et al., 2006*; *Mimori-Kiyosue et al., 2005*; *van der Vaart et al., 2013*). In this cell line, microtubules grow rapidly in the central parts of the cell, while at the cell margin, where CMSCs cluster in the vicinity of peripheral FAs, microtubule plus ends are tethered to the cortex and display persistent but slow growth due to the combined action of several types of microtubule regulators, including CLASPs, spectraplakins and KIF21A (*Drabek et al., 2006*; *Mimori-Kiyosue et al., 2005*; *van der Vaart et al., 2013*). This type of regulation prevents microtubule overgrowth at the cell edge and results in an orderly arrangement of microtubule plus ends perpendicular to the cell margin (*van der Vaart et al., 2013*) (*Figure 5A*). In cells with perturbed CMSCs, microtubule plus ends at the cell periphery become disorganized: the velocity of their growth at the cell margin increases, and their orientation becomes parallel instead of being perpendicular to the cell edge (*van der Vaart et al., 2013*) (*Figure 5A*).

Using live cell imaging of the microtubule plus end marker EB3-mRFP in KANK1/2 depleted cells rescued with the wild-type GFP-KANK1, we could indeed show that microtubule plus end growth velocity was almost 2.5 times slower at the cell margin compared to the central part of the cell, and the majority of microtubules at the cell margin grew at a 60–80° angle to the cell edge (*Figure 5B–E*). In the KANK1/2 depleted cells expressing KANK1 mutants, the velocity of microtubule growth in

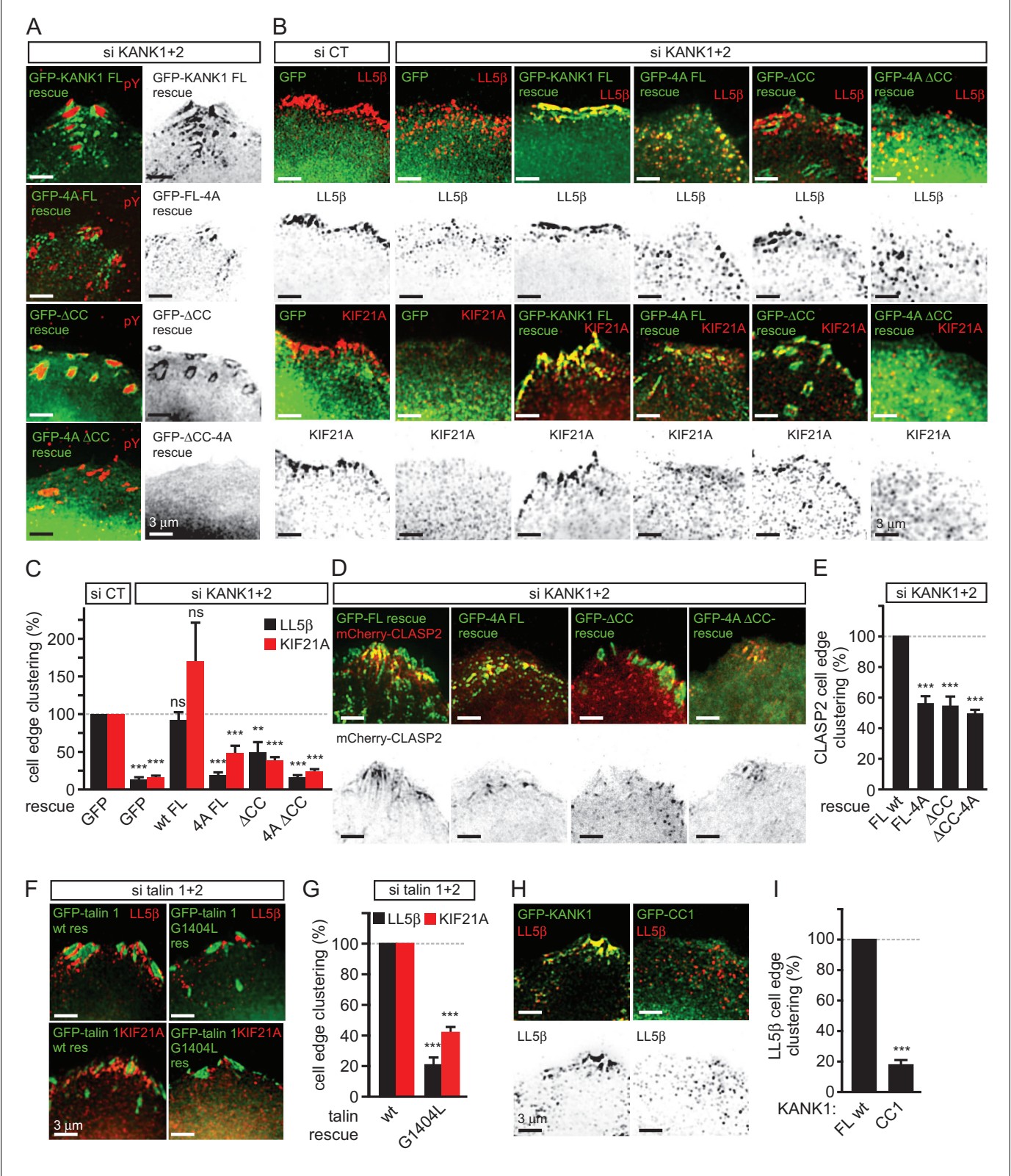

**Figure 4.** KANK1-talin interaction is required for recruiting CMSCs to FAs. (**A**) Widefield fluorescence images of HeLa cells depleted of KANK1 and KANK2 and expressing the indicated siRNA-resistant GFP-KANK1 fusions (rescue), stained for the FA marker phospho-tyrosine (pY). (**B**) Widefield fluorescence images of HeLa cells transfected with the control siRNA or siRNAs against KANK1 and KANK2, expressing GFP alone or the indicated siRNA-resistant GFP-KANK1 fusions and stained for LL5β or KIF21A. (**C**) Quantification of peripheral clustering of LL5β and KIF21A in cells treated as in

*Figure 4 continued on next page*

*Figure 4 continued*

panel (B) (n=12, 5–6 cells per condition). (D) TIRFM images of live HeLa cells depleted of KANK1 and KANK2 and co-expressing the indicated siRNA-resistant GFP-KANK1 fusions and mCherry-CLASP2. (E) Quantification of peripheral clustering of mCherry-CLASP2 in cells treated as in panel (D) (n=20, 8 cells per condition). (F) Widefield fluorescence images of HeLa cells transfected with the indicated GFP-KANK1 fusions and stained for endogenous LL5β. (G) Quantification of peripheral clustering of LL5β in cells treated as in panel (F) (n=12, 6 cells per condition). (H) Widefield fluorescence images of HeLa cells transfected with GFP-tagged KANK1 or its CC1 mutant and stained for LL5β. (I) Quantification of peripheral clustering of LL5β cells treated as in panel (H) (n=12, 6 cells per condition). Error bars, SEM; ns, non-significant; **p<0.005; ***p<0.001, Mann-Whitney U test.

The following source data is available for figure 4:

**Source data 1.** An Excel sheet with numerical data on the quantification of peripheral clustering of different markers represented as plots in *Figure 4C, E,G,I*.

central cell regions was not affected, but the growth rate at the cell periphery increased, and microtubules were growing at oblique angles to the cell margin (*Figure 5B–E*). The increase of the microtubule growth rate observed with the GFP-KANK1-ΔCC mutant was less strong than with the two 4A mutants (*Figure 5B–E*). This can be explained by the fact that GFP-KANK1-ΔCC was strongly clustered at FA rims (*Figure 3C*, *Figure 5B*), and, through its ankyrin repeat domain, could still recruit some KIF21A, a potent microtubule polymerization inhibitor (*van der Vaart et al., 2013*).

The results with rescue of talin1 and talin2 co-depletion with GFP-talin1 or its G1404L mutant fully supported the conclusions obtained with the KANK1-4A mutant: while in GFP-talin1-expressing cells microtubule growth at the cell edge was three fold slower than in the cell center, only a 1.5 fold difference was observed in GFP-talin1-G1404L expressing cells, and the proportion of microtubules growing parallel rather than perpendicular to the cell edge greatly increased (*Figure 5F–I*). We conclude that a single point mutation in talin1, which does not interfere with FA formation, is sufficient to perturb CMSC clustering and, as a consequence, induce microtubule disorganization in the vicinity of peripheral FAs.

## Discussion

In this study, we have shown that the conserved KN motif of KANK1 represents an LD-type ligand of talin, which allows this adaptor protein to accumulate in the vicinity of integrin-based adhesions. This function is likely to be conserved in the animal kingdom, as the KANK orthologue in *C. elegans*, Vab-19, in conjunction with integrins, plays important roles in a dynamic cell-extracellular matrix adhesion developmental process (*Ihara et al., 2011*). The exact impact of KANK1-talin binding likely depends on the specific system, as the loss of KANK proteins was shown to reduce motility of HeLa cells and podocytes (*Gee et al., 2015*; *Li et al., 2011*), but promote insulin-dependent cell migration in HEK293 cells (*Kakinuma et al., 2008*).

An important question is how KANK-talin1 binding mediates the localization of KANK1 to the rim but not the core of FAs. One possibility, suggested by our deletion analysis of KANK1, is that while the KN peptide alone can penetrate into FAs, larger KN-containing protein fragments are sterically excluded from the dense actin-containing core of the FA. However, our experiment with the KN-LacZ fusion did not support this simple idea, indicating that the underlying mechanism is likely to be more complex and might involve specific disordered or ordered domains and additional partners of KANK1, or other regulatory mechanisms. Interestingly, we found that reducing contractility with a ROCK1 inhibitor caused an increase in overlap of KANK1 with FA markers, suggesting that the interaction between KANK1 and talin might be mechanosensitive. An exciting possibility is that full length KANK1 can efficiently interact only with talin molecules at the periphery of focal adhesions because they are not fully incorporated into the focal adhesion structure and are thus less stretched. The KANK1 binding site on talin R7 overlaps with the high affinity actin binding site in talin (which spans R4-R8) (*Atherton et al., 2015*) and it is possible that different conformational populations of talin exist within adhesions and link to different cytoskeletal components.

Another important question is how KANK1 binding to the rim of focal adhesions can promote CMSC accumulation around these structures, a spatial arrangement in which most of the CMSC molecules cannot be in a direct contact with FAs. Previous work on CMSC complexes showed that they

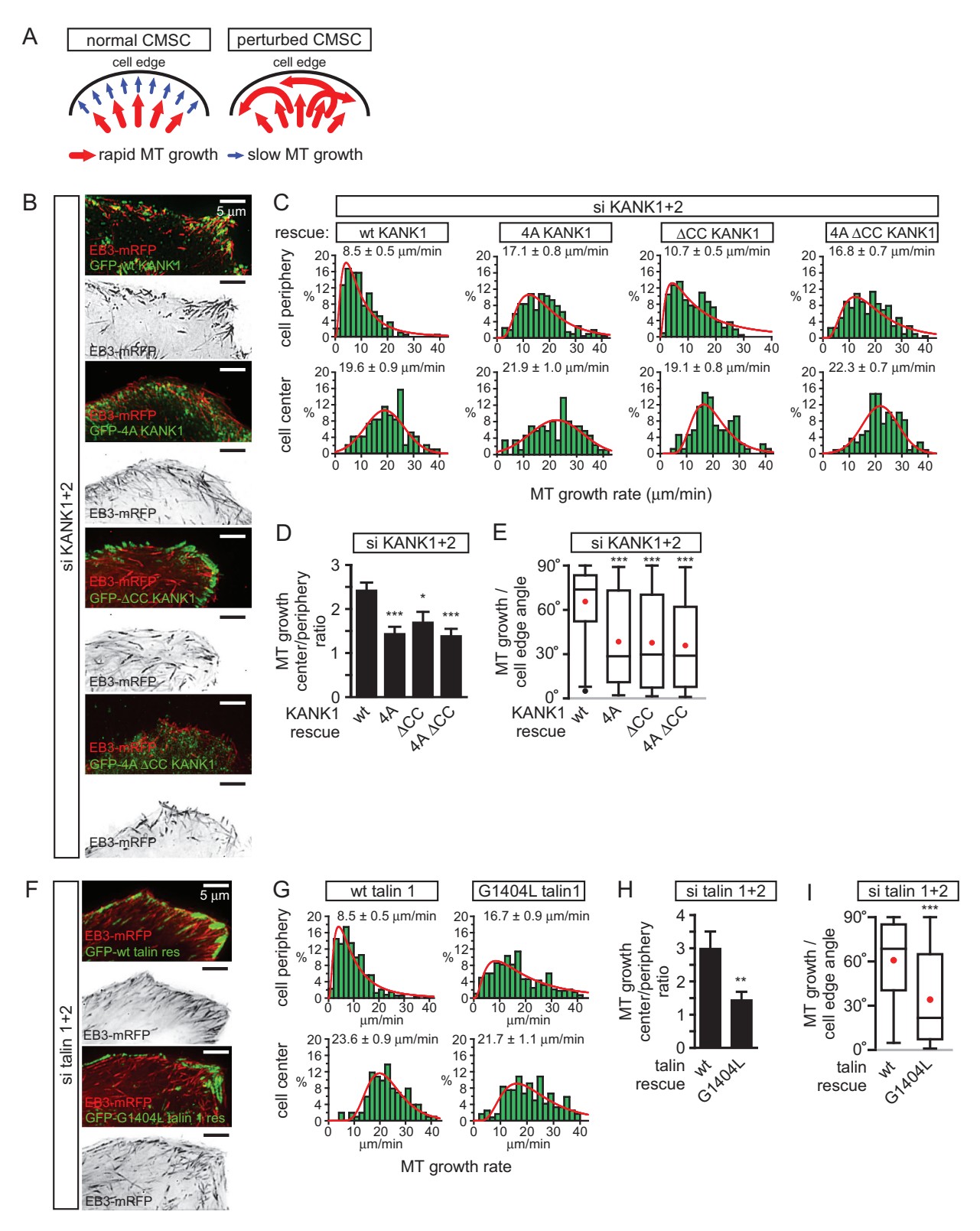

**Figure 5.** The role of talin-KANK1 interaction in regulating microtubule plus end dynamics around FAs. (**A**) Schematic representation of the pattern of microtubule growth in control HeLa cells and in cells with perturbed CMSCs, based on (*van der Vaart et al., 2013*). (**B**) TIRFM images of live HeLa cells depleted of KANK1 and KANK2 and co-expressing the indicated siRNA-resistant GFP-KANK1 fusions and EB3-mRFP. Images are maximum intensity projection of 241 images from time lapse recording of both fluorescence channels. (**C**) Distributions of microtubule growth rates at the 3 μm broad cell

*Figure 5 continued on next page*

*Figure 5 continued*

area adjacent to the cell edge, and in the central area of the ventral cortex for the cells treated as described in (B) (n=87–153, 7–8 cells). (D) Ratio of microtubule growth rate in the cell center and at the cell edge for the cells treated as described in B (n=7–8 cells). (E) Angles of microtubule growth relative to the cell margin for the cells treated as described in B. Box plots indicate the 25th percentile (bottom boundary), median (middle line), mean (red dots), 75th percentile (top boundary), nearest observations within 1.5 times the interquartile range (whiskers) and outliers (black dots) (n=93–114, 7–8 cells). (F) TIRFM images of live HeLa cells depleted of talin1 and talin 2 and co-expressing the indicated GFP-talin1 fusions and EB3-mRFP. Images are maximum intensity projection of 241 images from time lapse recordings of both fluorescence channels. (G) Distributions of microtubule growth rates at the 3 μm broad cell area adjacent to the cell edge, and in the central area of the ventral cortex for the cells treated as described in F (n=88–154, 7 cells). (H) The ratio of microtubule growth rate in the cell center and at the cell edge for the cells treated as described in panel (F) (n=7 cells). (I) Angles of microtubule growth relative to the cell margin for the cells treated as described in F. Box plots as in (E) (n=155–166, 10 cells). In all plots: error bars, SEM; ns, non-significant; **$p<0.01$; **$p<0.005$; ***$p<0.001$, Mann-Whitney U test.

The following source data is available for figure 5:

**Source data 1.** An Excel sheet with numerical data on the quantification of different aspects of microtubule organization and dynamics represented as plots in *Figure 5C–E,G–I*.

are formed through an intricate network of interactions. The 'core' components of these complexes, which can be recruited to the plasma membrane independently of each other, are LL5β (and in some cells, its homologue LL5α), liprins and KANKs (of which KANK1 seems to predominate in HeLa cells) (*Astro and de Curtis, 2015*; *Hotta et al., 2010*; *Lansbergen et al., 2006*; *van der Vaart et al., 2013*) (*Figure 6A*). The clustering of CMSC components is mutually dependent and relies on homo- and heterodimerization of liprins α1 and β1, the association between KANK1 and liprin-β1, the scaffolding protein ELKS, which binds to both LL5β and liprin-α1, and possibly additional interactions (*Astro and de Curtis, 2015*; *Lansbergen et al., 2006*; *van der Vaart et al., 2013*), while the microtubule-binding proteins, such as CLASPs and KIF21A, seem to associate as a second 'layer' with the membrane-bound CMSC-assemblies (*Figure 6A*). The CMSC 'patches' can remain relatively stable for tens of minutes, while their individual components are dynamic and exchange with different, characteristic turnover rates (*van der Vaart et al., 2013*).

The dynamic assemblies of CMSC components, which are spatially separate from other plasma membrane domains and which rely on multivalent protein-protein interactions, are reminiscent of cytoplasmic and nucleoplasmic membrane-unbounded organelles such as P granules and stress granules, the assembly of which has been proposed to be driven by phase transitions (*Astro and de Curtis, 2015*; *Brangwynne, 2013*; *Hyman and Simons, 2012*). The formation of such structures, which can be compared to liquid droplets, can be triggered by local concentration of CMSC components. It is tempting to speculate that by concentrating KANK1 at the FA rims, talin1 helps to 'nucleate' CMSC assembly, which can then propagate to form large structures surrounding FAs (*Figure 6B*). Additional membrane-bound cues, such as the presence of PIP3, to which LL5β can bind (*Paranavitane et al., 2003*), can further promote CMSC coalescence by increasing concentration of CMSC players in specific areas of the plasma membrane. This model helps to explain why the CMSC accumulation at the cell periphery is reduced but not abolished when PI3 kinase is inhibited (*Lansbergen et al., 2006*), and why the clustering of all CMSC components is mutually dependent. Most importantly, this model accounts for the mysterious ability of the two large and spatially distinct macromolecular assemblies, FAs and CMSCs, to form in close proximity of each other.

To conclude, our study revealed that a mechanosensitive integrin-associated adaptor talin not only participates in organizing the actin cytoskeleton but also directly triggers formation of a cortical microtubule-stabilizing macromolecular assembly, which surrounds adhesion sites and controls their formation and dynamics by regulating microtubule-dependent signaling and trafficking.

## Materials and methods

### Cell culture and transfection

HeLa Kyoto cell line was described previously (*Lansbergen et al., 2006*; *Mimori-Kiyosue et al., 2005*). HEK293T cells were purchased from ATCC; culture and transfection of DNA and siRNA into these cell lines was performed as previously described (*van der Vaart et al., 2013*). HaCaT cells

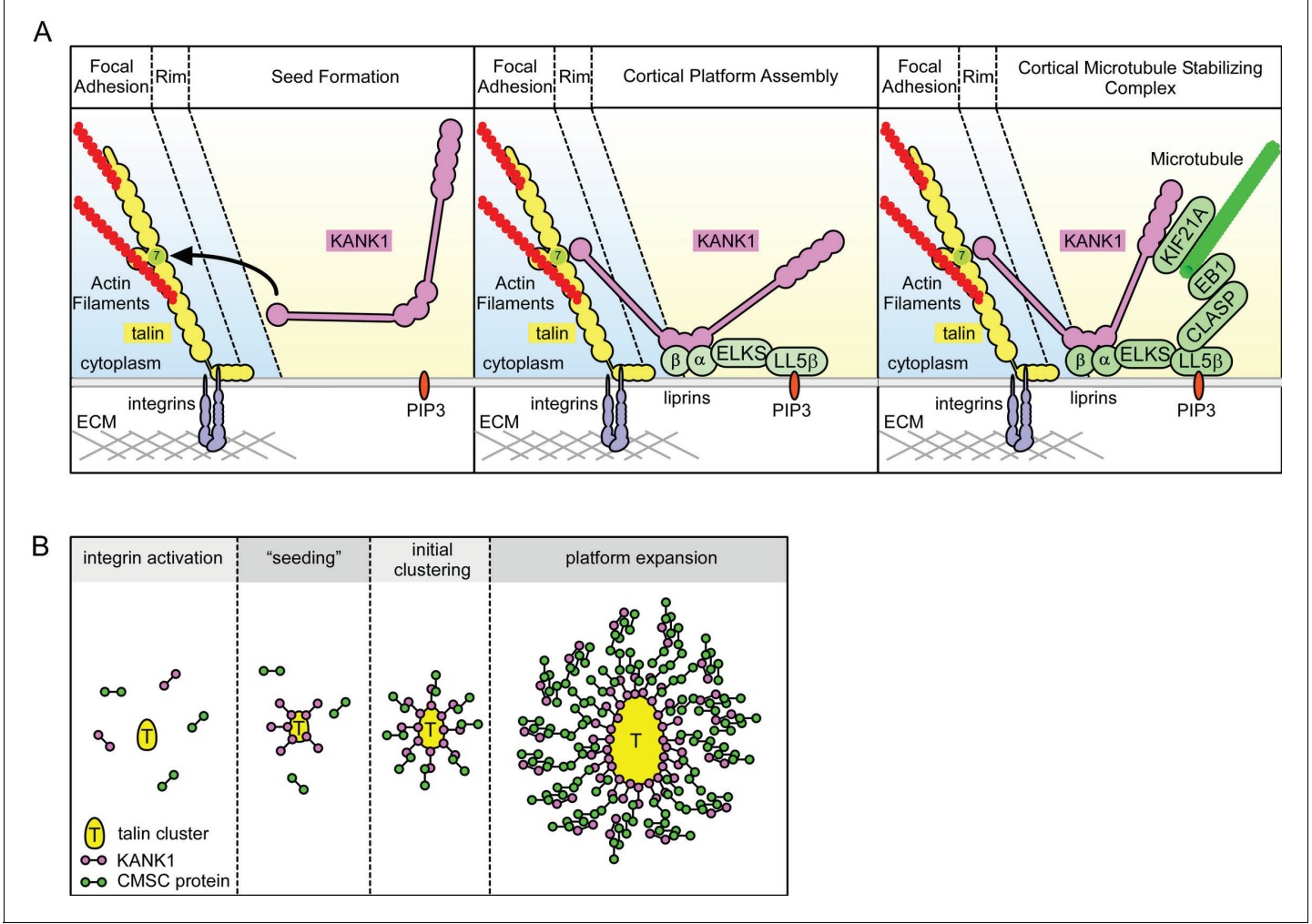

**Figure 6.** Model of talin-directed assembly of cortical microtubule stabilizing complex. (**A**) Three-step CMSC clustering around focal adhesion: *1)* KANK1 binds talin rod domain R7 via the KN motif, *2)* KANK1 initiates a cortical platform assembly by binding liprin-β1 via its CC1 domain, *3)* completion of CMSC assembly by further clustering of liprins, ELKS, LL5β, CLASP and KIF21A around FA. (**B**) KANK1 binding to nascent talin clusters acts as a 'seed' for macromolecular complex assembly and organization around a FA.

were purchased at Cell Line Service (Eppelheim, Germany) and cultured according to manufacturer's instructions. The cell lines were routinely checked for mycoplasma contamination using LT07-518 Mycoalert assay (Lonza, Switzerland).The identity of the cell lines was monitored by immunofluorescence-staining-based analysis with multiple markers. Blebbistatin was purchased from Enzo Life Sciences and used at 50 μM. Serum starvation in HeLa cells was done for 48 hr and focal adhesion assembly was stimulated by incubation with fetal calf serum-containing medium with or without blebbistatin for 2 hr. ROCK1 inhibitor Y-27632 was purchased at Sigma-Aldrich and used at 1 or 10 μM. Double stable HeLa cell line expressing GFP-KANK1 and TagRFP-paxillin was made by viral infection. We used a pLVIN-TagRFP-paxillin-based lentivirus and a pQC-GFP-KANK1-based retrovirus packaged in HEK293T cells using respectively Lenti-X HTX packaging and pCL-Ampho vectors. Antibiotic selection was applied to cells 48 hr after infection using 500 μg/ml G418 (Geneticin, Life Technologies) and 1 μg/ml puromycin (InvivoGen).

## DNA constructs and siRNAs

BioGFP-tagged KANK1 mutants were constructed using PCR and pBioGFP-C1 vector as previously described (*van der Vaart et al., 2013*). Rescue constructs for BioGFP-tagged KANK1 were either

based on the version previously described (*van der Vaart et al., 2013*) or a version obtained by PCR-based mutagenesis of the sequence AGTCAGCGTCTGCGAA to GGTGAGTGTGTGTGAG. mCherry-tagged paxillin construct was made by replacing GFP from pQC-GPXN (*Bouchet et al., 2011*) by mCherry (pmCherry-C1, Clontech). TagRFP-tagged paxillin construct was made by PCR-based amplification and cloning in pTagRFP-T-C1 (kind gift from Y. Mimori-Kiyosue, Riken Institute, Japan). HA-tagged KANK1 construct was generated by cloning KANK1 coding sequence into pMT2-SM-HA (gift of C. Hoogenraad, Utrecht University, The Netherlands). pLVX-IRES-Neo (pLVIN) vectors was constructed by cloning the IRES-neomycin resistance cassette from the pQCXIN plasmid (Clontech) into the pLVX-IRES-Puro plasmid (Clontech). The lentiviral Lenti-X HTX Packaging vector mix was purchased from Clontech. The retroviral packaging vector pCL-Ampho was kindly provided by E. Bindels, Erasmus MC, The Netherlands. The retroviral pQC-GFP-KANK1 vector was constructed by cloning GFP-KANK1 in pQCXIN and the lentiviral pLVIN-TagRFP-paxillin vector was constructed by cloning TagRFP-paxillin in pLVIN. BirA coding vector was described before (*van der Vaart et al., 2013*). GFP-tagged mouse talin 1 construct was a kind gift from Dr. A Huttenlocher (Addgene plasmid # 26724) (*Franco et al., 2004*). GFP-tagged KN-LacZ fusion was made using PCR-based amplification of KN and LacZ (kind gift, C. Hoogenraad, Utrecht University, The Netherlands), pBioGFP-C1 vector and Gibson Assembly mix (New England Biolabs). Site directed mutagenesis of KANK1 and talin1 constructs was realized by overlapping PCR-based strategy and validated by sequencing. mCherry-tagged CLASP2 construct was a gift from A. Aher (Utrecht University, The Netherlands). Single siRNAs were ordered from Sigma-Aldrich or Ambion, used at 5–15 nM, validated by Western blot analysis and/or immunofluorescence, and target sequences were the following: human KANK1 #1, CAGAGAAGGACATGCGGAT; human KANK1#2, GAAGTCAGCGTCTGCGAAA, human KANK2#1, ATGTCAACGTGCAAGATGA; human KANK2 #2, TCGAGAATCTCAGCACATA; human talin 1 #1, TCTGTACTGAGTAATAGCCAT; human talin 1 #2, TGAATGTCCTGTCAACTGCTG; human talin 2 #1, TTTCGTTTTCATCTACTCCTT; human talin 2 #2, TTCGTGTTTGGATTCGTCGAC. The combination of siRNAs talin1 #2 and talin2#1 was the most efficient and was used for the experiments shown in the paper.

## Pull down assays and mass spectrometry

Streptavidin-based pull down assays of biotinylated proteins expressed using pBioGFP-C1 constructs transfected in HEK293T cells was performed and analyzed as previously described (*van der Vaart et al., 2013*). For mass spectrometry sample preparation, streptavidin beads resulting from pull-downs assays were ran on a 12% Bis-Tris 1D SDS-PAGE gel (Biorad) for 1 cm and stained with colloidal coomassie dye G-250 (Gel Code Blue Stain Reagent, Thermo Scientific). The lane was cut and treated with 6.5 mM dithiothreitol (DTT) for 1 hr at 60°C for reduction and 54 mM iodoacetamide for 30 min for alkylation. The proteins were digested overnight with trypsin (Promega) at 37°C. The peptides were extracted with 100% acetonitrile (ACN) and dried in a vacuum concentrator. For RP-nanoLC-MS/MS, samples were resuspended in 10% formic acid (FA) / 5% DMSO and was analyzed using a Proxeon Easy-nLC100 (Thermo Scientific) connected to an Orbitrap Q-Exactive mass spectrometer. Samples were first trapped (Dr Maisch Reprosil C18, 3 µm, 2 cm × 100 µm) before being separated on an analytical column (Agilent Zorbax 1.8 µm SB-C18, 40 cm × 50 µm), using a gradient of 180 min at a column flow of 150 nl min$^{-1}$. Trapping was performed at 8 µL/min for 10 min in solvent A (0.1 M acetic acid in water) and the gradient was as follows 15- 40% solvent B (0.1 M acetic acid in acetonitrile) in 151 min, 40–100% in 3 min, 100% solvent B for 2 min, and 100% solvent A for 13 min. Nanospray was performed at 1.7 kV using a fused silica capillary that was pulled in-house and coated with gold (o.d. 360 µm; i.d. 20 µm; tip i.d. 10 µm). The mass spectrometers were used in a data-dependent mode, which automatically switched between MS and MS/MS. Full scan MS spectra from m/z 350 – 1500 were acquired at a resolution of 35.000 at m/z 400 after the accumulation to a target value of 3E6. Up to 20 most intense precursor ions were selected for fragmentation. HCD fragmentation was performed at normalized collision energy of 25% after the accumulation to a target value of 5E4. MS2 was acquired at a resolution of 17,500 and dynamic exclusion was enabled. For data analysis, raw files were processed using Proteome Discoverer 1.4 (version 1.4.1.14, Thermo Scientific, Bremen, Germany). Database search was performed using the swiss-prot human database (version 29th of January 2015) and Mascot (version 2.5.1, Matrix Science, UK) as the search engine. Carbamidomethylation of cysteines was set as a fixed modification and oxidation of methionine was set as a variable modification. Trypsin was specified as enzyme and up to two

miss cleavages were allowed. Data filtering was performed using a percolator (*Käll et al., 2007*), resulting in 1% false discovery rate (FDR). Additional filter was ion score >20.

## Antibodies and immunofluorescence cell staining

Antibodies against HA and GFP tags, and liprin β1 used for Western blot analysis were previously described (*van der Vaart et al., 2013*). Rabbit antibodies against KANK1 (HPA005539) and KANK2 (HPA015643) were purchased at Sigma-Aldrich. Western blot analysis of KANK1 was performed using rabbit polyclonal KANK1 antibody (A301-882A) purchased at Bethyl Laboratories. Talin immunofluorescence staining was performed using mouse monoclonal 8d4 antibody (Sigma-Aldrich). Western blot analysis of talin 1 and 2 expression was performed using respectively the isotype specific mouse monoclonal 97H6 (Sigma-Aldrich) and 68E7 (Abcam) antibodies. Ku80 (7/Ku80) antibody was purchased from BD Biosciences. Immunofluorescence staining of KANK1, LL5β, liprin β1, KIF21A and CLASP2 in HeLa and HaCaT cells was performed using the antibodies and procedures previously described (*Lansbergen et al., 2006*; *van der Vaart et al., 2013*). F-actin was stained using Alexa Fluor 594-conjugated phalloidin (Life Technologies). Phospho-tyrosine mouse antibody (PT-66) was purchased from Sigma-Aldrich and rabbit FAK phospho-tyrosine 397 was purchased from Biosource.

## Microscopy and analysis

Fixed samples and corresponding immunofluorescence images were acquired using widefield fluorescence illumination on a Nikon Eclipse 80i or Ni upright microscope equipped with a CoolSNAP HQ2 CCD camera (Photometrics) or a DS-Qi2 camera (Nikon) an Intensilight C-HGFI precentered fiber illuminator (Nikon), ET-DAPI, ET-EGFP and ET-mCherry filters (Chroma), Nikon NIS Br software, Plan Apo VC 100x NA 1.4 oil, Plan Apo Lambda 100X oil NA 1.45 and Plan Apo VC 60x NA 1.4 oil (Nikon) objectives. TIRFM-based live cell imaging was performed using the setup described before (*van der Vaart et al., 2013*) or a similar Nikon Ti microscope-based Ilas$^2$ system (Roper Scientific, Evry, FRANCE) equipped with dual laser illuminator for azimuthal spinning TIRF (or Hilo) illumination, 150 mW 488 nm laser and 100 mW 561 nm laser, 49,002 and 49,008 Chroma filter sets, EMCCD Evolve mono FW DELTA 512x512 camera (Roper Scientific) with the intermediate lens 2.5X (Nikon C mount adapter 2.5X), CCD camera CoolSNAP MYO M-USB-14-AC (Roper Scientific) and controlled with MetaMorph 7.8.8 software (Molecular Devices). Simultaneous imaging of green TIRFM imaging was performed as described before (*van der Vaart et al., 2013*) or using the Optosplit III image splitter device (Andor) on the Ilas$^2$ system.

For presentation, images were adjusted for brightness and processed by Gaussian blur and Unsharp mask filter using ImageJ 1.47v (NIH). Fluorescence profiles are values measured by line scan analysis in ImageJ, normalized by background average fluorescence, expressed as a factor of the baseline value obtained for individual channel and plotted as a function of maximum length factor of the selection line (distance ratio). Protein clustering at the cell edge is the ratio of the total fluorescence in the first 5 μm from the cell edge to the next 5 μm measured by line scan analysis in ImageJ after thresholding for cell outline marking and out-of-cell region value assigned to zero. The results were plotted as percentage of control condition average value. FA counting and area measurement was performed using Analyze Particles under ImageJ on focal adhesion binary mask obtained after Gaussian blur/threshold-based cell outline marking and background subtraction (rolling ball radius, 10 pixels). KANK1/talin colocalization was analyzed using Pearson R value provided by Colocalization Analysis plugin under FiJi-ImageJ and a 3 μm diameter circular ROI centered on talin clusters detected by immunofluorescent staining.

nEB3-mRFP dynamics was recorded by 0.5 s interval time lapse TIRF imaging. Microtubule growth was measured using kymographs obtained from EB3-mRFP time lapse image series, plotted and presented as previously described (*van der Vaart et al., 2013*). Ratio of microtubule growth in cell center to periphery was obtained as values for individual cells. Microtubule growth trajectory angle to the cell edge was manually measured in ImageJ using tracks obtained by maximum intensity projection of EB3-mRFP image series.

## Expression of recombinant talin polypeptides

The cDNAs encoding murine talin1 residues 1357–1653 (R7-R8), 1357–1653 Δ1454–1586 (R7) and 1461–1580 (R8) were synthesized by PCR using a mouse talin1 cDNA as template and cloned into

the expression vector pet151-TOPO (Invitrogen) (*Gingras et al., 2010*). Talin mutants were synthesized by GeneArt. Talin polypeptides were expressed in *E. coli* BL21(DE3) cultured either in LB for unlabeled protein, or in M9 minimal medium for the preparation of isotopically labeled samples for NMR. Recombinant His-tagged talin polypeptides were purified by nickel-affinity chromatography following standard procedures. The His-tag was removed by cleavage with AcTEV protease (Invitrogen), and the proteins were further purified by anion-exchange. Protein concentrations were determined using their respective extinction coefficient at 280 nm.

## Fluorescence polarization assays

KANK peptides with a C-terminal cysteine residue were synthesized by Biomatik (USA):

KANK1(30–60)C – PYFVETPYGFQLDLDFVKYVDDIQKGNTIKKC

KANK1(30–68)C - PYFVETPYGFQLDLDFVKYVDDIQKGNTIKKLNIQKRRKC

KANK1-4A - PYFVETPYGFQAAAAFVKYVDDIQKGNTIKKLNIQKRRKC

KANK2(31–61)C - PYSVETPYGYRLDLDFLKYVDDIEKGHTLRRC

Fluorescence Polarization was carried out on KANK peptides with a carboxy terminal cysteine. Peptide stock solutions were made in PBS (137 mM NaCl, 27 mM KCl, 100 mM $Na_2HPO_4$, 18 mM $KH_2PO_4$), 100 mg/ml TCEP and 0.05% Triton X-100, and coupled via the carboxy terminal cysteine to the Thiol reactive BIODIPY TMR dye (Invitrogen). Uncoupled dye was removed by gel filtration using a PD-10 column (GE Healthcare). The labeled peptide was concentrated to a final concentration of 1 mM using a centricon with 3K molecular weight cut off (Millipore).

The Fluorescence Polarization assay was carried out on a black 96well plate (Nunc). Titrations were performed in triplicate using a fixed 0.5 µM concentration of peptide and an increasing concentration of Talin R7-R8 protein within a final volume of 100 µl of assay buffer (PBS). Fluorescence Polarization measurements were recorded on a BMGLabTech CLARIOstar plate reader at room temperature and analyzed using GraphPad Prism (version 6.07). $K_d$ values were calculated with a nonlinear curve fitting using a one site total and non-specific binding model.

## NMR spectroscopy

NMR experiments for the resonance assignment of talin1 R7, residues 1357–1653 Δ1454–1586 were carried out with 1 mM protein in 20 mM sodium phosphate, pH 6.5, 50 mM NaCl, 2 mM dithiothreitol, 10% (v/v) 2H2O. NMR spectra were obtained at 298 K using a Bruker AVANCE III spectrometer equipped with CryoProbe. Proton chemical shifts were referenced to external 2,2-dimethyl-2-silapentane- 5-sulfonic acid, and $^{15}N$ and $^{13}C$ chemical shifts were referenced indirectly using recommended gyromagnetic ratios (*Wishart et al., 1995*). The spectra were processed using Topspin and analyzed using CCPN Analysis (*Skinner et al., 2015*). Three-dimensional HNCO, HN(CA)CO, HNCA, HN(CO)CA, HNCACB, and CBCA(CO)NH experiments were used for the sequential assignment of the backbone NH, N, CO, CA, and CB resonances.

The backbone resonance assignments of mouse talin1 R7 (1357–1653 Δ1454–1586) have been deposited in the BioMagResBank with the accession number 19139.

## Acknowledgements

We thank Y Mimori-Kiyosue (Riken Institute, Japan), A Huttenlocher (University of Wisconsin), A Aher and C Hoogenraad (Utrecht University, The Netherlands) for the gift of reagents. We thank M Geleijnse and A Floor for help with siRNA validation, cell culture and fluorescent staining. We are grateful to I Grigoriev (Utrecht University, The Netherlands) for assistance with microscopy and advice about microtubule dynamics analysis, and to JD Kaiser for advice about PowerPoint. This work was supported by the Netherlands organization for Scientific Research (NWO) ALW VICI grant 865.08.002 and a European Research Council (ERC) Synergy grant 609822 to AA, a BBSRC grant (BB/N007336/1) to BTG, Human Frontier Science Program RGP00001/2016 grant to AA and BTG, NWO VIDI grant (723.012.102) for AFMA and as part of the NWO National Roadmap Large-scale Research Facilities of the Netherlands (project number 184.032.201) for AFMA, AJRH and HP, and Fondation pour la Recherche Médicale and Marie Curie International Intra-European Fellowship to BPB Y-C A is supported by the MARIE SKŁODOWSKA-CURIE ACTIONS Innovative Training Network (ITN) 675407 PolarNet.

## Additional information

### Competing interests

AA: Reviewing editor, *eLife*. The other authors declare that no competing interests exist.

### Funding

| Funder | Grant reference number | Author |
|---|---|---|
| Fondation pour la Recherche Médicale (FRM) | postdoctoral fellowship | Benjamin P Bouchet |
| Marie Curie International Intra-European Fellowship | postdoctoral fellowship | Benjamin P Bouchet |
| Nederlandse Organisatie voor Wetenschappelijk Onderzoek | National Roadmap for Large-Scale facilities Project 184.032.201 | AF Maarten Altelaar Albert JR Heck |
| Nederlandse Organisatie voor Wetenschappelijk Onderzoek | VIDI 723.012.102 | AF Maarten Altelaar |
| Biotechnology and Biological Sciences Research Council | Grant BB/N007336/1 | Benjamin T Goult |
| Human Frontier Science Program | RGP00001/2016 | Benjamin T Goult Anna Akhmanova |
| Nederlandse Organisatie voor Wetenschappelijk Onderzoek | ALW VICI 865.08.002 | Anna Akhmanova |
| European Research Council | ERC Synergy 609822 | Anna Akhmanova |
| Marie Curie Actions Innovative Training Network | ITN PolarNet 675407 | Anna Akhmanova |

The funders had no role in study design, data collection and interpretation, or the decision to submit the work for publication.

### Author contributions

BPB, BTG, Conception and design, Acquisition of data, Analysis and interpretation of data, Drafting or revising the article; REG, Y-CA, DvdW, HP, Acquisition of data, Analysis and interpretation of data, Drafting or revising the article; GJ, Conception and design, Drafting or revising the article, Contributed unpublished essential data or reagents; AFMA, Analysis and interpretation of data, Drafting or revising the article; AJRH, AA, Conception and design, Analysis and interpretation of data, Drafting or revising the article

### Author ORCIDs

Albert JR Heck, http://orcid.org/0000-0002-2405-4404
Benjamin T Goult, http://orcid.org/0000-0002-3438-2807
Anna Akhmanova, http://orcid.org/0000-0002-9048-8614

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
