## [Decision Letter]

Thank you for submitting your article "Talin-KANK1 interaction controls the recruitment of cortical microtubule stabilizing complexes to focal adhesions" for consideration by *eLife*. Your article has been reviewed by three peer reviewers, one of whom is a member of our Board of Reviewing Editors and the evaluation has been overseen by Vivek Malhotra as the Senior Editor. The following individuals involved in review of your submission have agreed to reveal their identity: Terry Lechler (Reviewer #2).

The reviewers have discussed the reviews with one another and the Reviewing Editor has drafted this decision to help you prepare a revised submission.

Summary:

This manuscript presents a molecular model for the connection between focal adhesions and associated complexes that stabilize microtubules (called cortical microtubule stabilization complexes, CMSC). KANK1 and 2 are able to bind to both talin at focal adhesions as well as liprins and KIF21A in CMSC's. In the absence of this bridging interaction, CMSCs do not form around focal adhesions. Important strengths of this work include the rational design of point mutants that disrupted talin-KANK interactions and the careful analysis of localizations in backgrounds in which either talins or KANKs were depleted. Functionally, disrupting the association between talin and KANK results in CMSC disruption, which perturbs microtubule growth speeds and organization at the cell cortex. Thought provoking models for the assembly of CMSCs around focal adhesions are presented in the Discussion. Overall, all 3 reviewers felt that this is a very well executed manuscript that provides substantial new insight into how focal adhesions locally control microtubule dynamics.

Essential revisions:

1) Based on the results of blebbistatin treatment (Results section) the authors concluded that 'the organization of CMSCs at the cell cortex might be controlled by tension-sensitive components of FAs'. However, it is obvious from the image of the blebbistatin-treated cell (Figure 1—figure supplement 2) that this treatment caused complete disassembly of the focal adhesions rather than moderate decrease in actomyosin contractility. To support author's statement, which is interesting and important, an additional experiment is required. The authors should attenuate myosin contractility by either plating the cells on a soft substrate (similar to the experiments in Myers et al., 2011) or by treating the cells with low concentration of blebbistatin or Y-27632 (see Oakes et al., 2012). These treatments will decrease mechanical tension across force-bearing adhesion molecules without compromising structural integrity and signaling function of the focal adhesions. Finding that low contractility impairs CMSC organization will significantly strengthen the paper.

Additional experiments to consider:

1) It would improve the impact of the manuscript to demonstrate whether stabilization of microtubules by KANK1-talin complex has an effect on focal adhesion dynamics. In the introduction section of the manuscript the authors provided several lines of evidence demonstrating the crosstalk between microtubules and focal adhesions. However, the experimental data presented in the manuscript (Figure 3; Figure 4) show very little, if any, difference in the focal adhesion size upon perturbing talin-KANK interaction.

2) Since MT localization at FAs is important in cell polarization and migration, it would seem useful to test whether key mutant rescues of talin1 or KANK1 knockdowns have normal polarization and migration.

---

## [Author Response]

*Essential revisions:*

*1) Based on the results of blebbistatin treatment (Results section) the authors concluded that 'the organization of CMSCs at the cell cortex might be controlled by tension-sensitive components of FAs'. However, it is obvious from the image of the blebbistatin-treated cell (Figure 1—figure supplement 2) that this treatment caused complete disassembly of the focal adhesions rather than moderate decrease in actomyosin contractility. To support author's statement, which is interesting and important, an additional experiment is required. The authors should attenuate myosin contractility by either plating the cells on a soft substrate (similar to the experiments in Myers et al., 2011) or by treating the cells with low concentration of blebbistatin or Y-27632 (see Oakes et al., 2012). These treatments will decrease mechanical tension across force-bearing adhesion molecules without compromising structural integrity and signaling function of the focal adhesions. Finding that low contractility impairs CMSC organization will significantly strengthen the paper.*

We thank the reviewers for this excellent suggestion. We have treated the cells with a low dose of the ROCK1 inhibitor Y-27632 and imaged focal adhesions and KANK1 in both live and fixed cells. As shown in the new Figure 1—figure supplement 2 and Supplemental Video S1, we found that this treatment induced a reduction in focal adhesion size, as expected. Interestingly this treatment led to an increased the penetration of KANK1 into focal adhesions and thus enhanced the colocalization between KANK1 and focal adhesion markers; subsequently, gradual dispersion of KANK1 clusters was observed. These data strongly suggest that the interaction between KANK1 and talin is mechanosensitive and can be regulated by tension at focal adhesions. These data are incorporated into the Results.

*Additional experiments to consider:*

1) It would improve the impact of the manuscript to demonstrate whether stabilization of microtubules by KANK1-talin complex has an effect on focal adhesion dynamics. In the introduction section of the manuscript the authors provided several lines of evidence demonstrating the crosstalk between microtubules and focal adhesions. However, the experimental data presented in the manuscript (Figure 3; Figure 4) show very little, if any, difference in the focal adhesion size upon perturbing talin-KANK interaction.

We have now measured focal adhesion size in HeLa cells where both talin1 and talin2 were depleted and adhesion was rescued by either the wild type talin1 or the G1404L mutant. We found no significant differences between focal adhesion sizes in these two conditions, although cell spreading was mildly increased in cells transfected with the G1404L mutant (see Figure 3—figure supplement 1 and the Results of the revised manuscript (subsection “Talin1-KANK1 interaction controls cortical organization of CMSC components“). We note that the HeLa cells used here migrate extremely slowly. As a result, they have a highly symmetric shape and represent an excellent model to study microtubule organization and dynamics but are not a good model for investigation of focal adhesion remodeling during cell migration. To address the 2 consequences of disrupting KANK1-talin interaction on focal adhesion turnover, one would therefore need to move to another cell system, such as, for example, 3T3 fibroblasts or HaCaT keratinocytes, in which the cortical microtubule stabilization complexes strongly associate with focal adhesions at the leading cell edge (Lansbergen et al., 2006 Dev Cell, Stehbens et al., 2014 Nat Cell Biol). Unfortunately, these cell models are difficult to transfect, and thus establishment of multiple stable cell lines will be needed to properly address the impact of perturbation of KANK1-talin binding on cell migration and focal adhesion turnover. Such experiments would require at least several months to perform, and, in our opinion, go beyond the scope of the current paper but should constitute a separate study.

*2) Since MT localization at FAs is important in cell polarization and migration, it would seem useful to test whether key mutant rescues of talin1 or KANK1 knockdowns have normal polarization and migration.*

As indicated above, we think that investigation of the impact of disruption of KANK1-talin interaction on cell polarity and migration would require the use of another model of cell motility. Development of proper tools in such a new model will take multiple months, and thus, in our opinion, goes beyond the scope of this manuscript.